# Artificial Neural Network Detects Hip Muscle Forces as Determinant for Harmonic Walking in People after Stroke

**DOI:** 10.3390/s22041374

**Published:** 2022-02-11

**Authors:** Marco Iosa, Maria Grazia Benedetti, Gabriella Antonucci, Stefano Paolucci, Giovanni Morone

**Affiliations:** 1Department of Psychology, Sapienza University of Rome, 00185 Rome, Italy; gabriella.antonucci@uniroma1.it; 2IRCCS Fondazione Santa Lucia, 00179 Rome, Italy; s.paolucci@hsantalucia.it (S.P.); g.morone@hsantalucia.it (G.M.); 3Physical Medicine and Rehabilitation Unit, IRCCS-Istituto Ortopedico Rizzoli, 40136 Bologna, Italy; benedetti@ior.it

**Keywords:** artificial intelligence, gait analysis, gait, golden ratio, quadriceps, iliopsoas

## Abstract

Many recent studies have highlighted that the harmony of physiological walking is based on a specific proportion between the durations of the phases of the gait cycle. When this proportion is close to the so-called golden ratio (about 1.618), the gait cycle assumes an autosimilar fractal structure. In stroke patients this harmony is altered, but it is unclear which factor is associated with the ratios between gait phases because these relationships are probably not linear. We used an artificial neural network to determine the weights associable to each factor for determining the ratio between gait phases and hence the harmony of walking. As expected, the gait ratio obtained as the ratio between stride duration and stance duration was found to be associated with walking speed and stride length, but also with hip muscle forces. These muscles could be important for exploiting the recovery of energy typical of the pendular mechanism of walking. Our study also highlighted that the results of an artificial neural network should be associated with a reliability analysis, being a non-deterministic approach. A good level of reliability was found for the findings of our study.

## 1. Introduction

Strokes are one of the most important leading causes of movement disability in Europe and the US; each year approximately 780,000 people experience a new (77%) or recurrent (13%) stroke, with 30–35 deaths for every 100,000 people [1]. About 50% of stroke patients leave the rehabilitation hospital in a wheelchair, less than 15% are able to walk indoor without aids, less than 10% are able to walk outdoor, and less than 5% are able to climb stairs [2]. In most of the patients who recovered their walking ability, some deficits are still evident: gait is slow, asymmetric, instable and disharmonic [3]. Instrumented gait analysis, based on stereophotogrammetry, force platforms, electromyography, and occasionally metabolimeters and/or inertial wearable devices can allow the characterization of stroke gait patterns through a huge quantity of kinematic and kinetic data, which can be added to clinical data [4,5,6]. Recently, it has been suggested that Artificial Intelligence could be used to manage this large amount of data. Artificial neural networks (ANN) are sets of non-linear data computational models consisting of input and output layers as well as one or more hidden layers (Figure 1). The connections between neurons in each layer have associated weights, which are iteratively adjusted by the training algorithm to minimize error and provide accurate predictions [7]. For stroke patients, artificial neural networks have been used as models for screening [8], risk identification [9] or as a prognostic tool [10], especially when the possible relationships among factors are not linear. This is the case for harmony in walking, a feature that can be assessed, according to suggestions, by the ratio between stride and stance [11]. In fact, it was found that, in healthy subjects, when this ratio is close to a specific number—the so-called golden ratio (GR, about 1.618…)—the gait cycle assumes a particular autosimilar harmonic structure (Figure 2) [11]. This proportion derives from the ancient geometrical problem, cited by Euclid, to cut a segment so that the length of the whole segment stands to that of the longer part, as well as that of longer part standing to that of shorter one [12]. The problem is solved by a cut dividing the segment in a proportion equal to 61.8%–38.2%, which is very close to the division in stance and swing phases of the physiological gait cycle in healthy subjects. When this proportion is respected, not only the proportion between the duration of the stride and that of the stance are equally to the GR, but also the proportion between the stance and the swing, and even that between the swing and the sum of the two double support phases for the property of autosimilarity of the GR [11]. This number was already found at the basis of many biological and anthropic spatial and temporal harmonic structures [12]. The stride to stance ratio was found to be directly altered in the case of Parkinson’s Disease [13] and cerebellar ataxia [14], whereas its role in stroke has still been unclear, when the deficit seemed to directly reduce walking speed and, in turn, to indirectly affect this ratio [3]. However, this parameter has an optimal value (when the stance is 61.8% of the stride), and both reductions and increments with respect to this value decrease the harmony of walking. This limits the use of conventional computational linear models (such as regressions) to identify the factors that mainly affect this parameter, because the relationships of other factors to the gait ratio could be non-linear. Among the non-linear approaches, a new recent and promising approach is the use of artificial intelligence. Lee and colleagues were pioneers in using an ANN with an accuracy of approximately 80% in identifying movement disorders from spatial parameters obtained by video analysis of gait [15]. Scheffer and Cloete had the intuition of the potentialities of combining two emerging technologies: artificial neural networks and motion capture [16]. Their results suggested the usability of ANN and gait analysis for planning gait rehabilitation therapy and monitoring its outcomes in the case of a stroke.

Post-stroke individuals often exhibit alterations in lower limb kinematics during walking, with an increased pelvic obliquity and hip abduction coupled with reduced knee flexion, behaviors that could be related to an abnormal cross-planar coupling of muscle activity [17]. Energy consumption during walking is strictly related to the mechanical work [18], and to conserve energy, stroke patients often use the paretic lower-limb as a rigid shaft while the non-paretic limb puts in action compensatory strategies [19]. This energy consumption seems to be minimized when patients walk in the golden proportion [14], however the stride to stance ratio (gait ratio) has received poor attention in previous studies.

The aim of this study was to investigate the factors associated with gait ratio, analyzing data by means of an artificial neural network. Demographical, clinical, and instrumented data have been collected and used as inputs for the neural network, as shown in Figure 1.

## 2. Materials and Methods

### 2.1. Participants and Clinical Assessment

Twenty-six stroke patients were enrolled into the study. The mean age of this sample was 69.7 ± 7.4 years (16 males and 10 females, 16 with left body hemiparesis and 10 with right hemiparesis). This sample is characterized by a Barthel Index score of 72 ± 16 (that is a measure of independency in activities of daily living going from 0, denoting total dependency, up to 100, denoting total independence). A Nottingham Extended ADL (activities of daily living) Scale of 11 ± 8, VAS score was used to assess pain of 20 ± 25 (Visual Analogue Scale going from 0, denoting no pain, up to 100, denoting insupportable pain). Patients performed the six minute walking test (6 MWT), covering an average distance of 109 ± 54 m (16 patients without any stop, the other 10 patients with a number of stop for resting going from 1 up to 7). The mean Physiological Cost Index (PCI) of walking was 1 ± 1, and it was assessed as the difference between the heart frequency measured at the end of six minute walking and that at initial rest, divided by the average walking speed [20].

The range of joint motion was measured using a goniometer for hip flexion, hip abduction, hip adduction, knee flexion, ankle dorsiflexion and ankle plantaflexion of the plegic body side. A dynamometer was used to assess the force of quadriceps and ileo-psoas muscles, bilaterally. These muscles have been chosen because they are the two active during the foot off that defines the end of the stance [4] and hence are hypothesized as those that could mainly affect the stride to stance ratio.

A total of 33 parameters for each one of the 26 patients formed the input data of the artificial neural network, resulting in 858 values being entered in the input layers. As shown in Figure 1, the 33 parameters were related to demographical data (2: age, gender), clinical scale scores (3: pain Visual Analogue Scale, Barthel Index, Nottingham Extended ADL Scale), hip muscle forces (4: maximum force of iliopsoas and quadriceps of both side), ranges of joint motion (6: hip flexion, abduction and adduction, knee flexion, ankle plantaflexion of paretic side), 6 MWT (walked distance, number of stops, pain during the test) cardiological data (7: heart rate, systolic and diastolic pressure at rest and after the 6 MWT, Physiological Cost Index), and gait data (7: stride length, stride duration, cadence of paretic and non-paretic steps, speed of paretic and non-paretic steps, compensation strategies).

### 2.2. Gait Analysis

Gait analysis was performed using a stereophotogrammetric system with eight infrared cameras (Vicon 612, Oxford Metrics, Oxford, UK), a device guaranteeing good to excellent reproducibility for stroke patients and considered to be the gold standard for human movement analysis [21]. Twenty-two retroflective markers were attached to the skin of each patient with bioadhesive tape on the body landmarks of the IOR (Istituto Ortopedico Rizzoli) gait protocol [22]. After the anatomical measurements of body segment length and width, the patient was asked to be firm in a standing position for 5 s to allow a registration of posture. After that, the patient was asked to walk at a comfortable speed. The data were averaged among at least three walking trials, and hence at least six gait cycles for each limb to obtain the following spatiotemporal gait parameters: walking speed (the distance covered in a unit of time), cadence (number of strides per minute), stride length (obtained as the sum of the lengths of two consecutive steps, one performed with the paretic leg and one with the non-paretic leg). The gait ratio was calculated as the ratio between stride duration and stance duration of the paretic side, and its value was compared with the estimated output of the ANN.

### 2.3. Neural Network Analysis

ARIANNA is an ARtificial Intelligence Assistant for Neural Network Analysis already developed and used in previous studies on the movement analysis of stroke patients [23,24]. ARIANNA is a Feed Forward Neural Network Multilayer Perceptron formed by four layers: the input layer, from which entered the 33 variables related to demographical, clinical, tests, and instrumented gait analysis; two hidden layers of 5 elements each; and a final output layer related to the gait ratio (Figure 1). Data travel in only one direction, from the input nodes through the two hidden layers to the output nodes [24]. The activation function for all units in the hidden layers and that for the output layer were both a hyperbolic tangent. The chosen computational procedure was based on an online training (details: initial learning = 1.2; lower learning = 0.001, learning epochs = 10, momentum = 0.9, interval center = 0, interval offset = ±0.5, memsize = 1000, steps without error = 1, error change = 0.0001, error ratio = 0.001). A synaptic weight was associated with each connection of the neural network: this coefficient estimates the relationship between the units in a given layer to the units in the following layer. The synaptic weights were then normalized as the percentage of the maximum weight and considered as an indicator of the importance of each specific input parameter to determine the output parameter. ARIANNA was developed using the IBM SPSS Neural Networks module of IBM SPSS Statistic, version 23.

### 2.4. Statistical Analysis

Sample size was computed on the basis of previous studies on gait analysis in stroke patients [3,23]. The importance associated by ANN with each factor are reported in terms of percentage of the maximum importance. Data are reported as mean and standard deviation. To assess the reliability of the results of the ANN, the computation was repeated 10 times, the data were averaged, and a coefficient of variation (that is, the percentage ratio between standard deviation and average value) was measured for each parameter. Bioexponential fits were used to model the relationships between two parameters, reporting the coefficient of determination R^2^ to assess the goodness of fits. All these analyses were performed using MATLAB R2016b, MathWorks Inc. (Natick, MA, USA).

## 3. Results

The neural network analysis associated a percentage weight with each parameter to predict the value of the gait ratio; the results were mainly associated with some spatio-temporal parameter of walking, such as paretic stride length (88.9 ± 8.8%), walking speed (86.8 ± 9.9%), stride duration (85.7 ± 11.0%), or cadence (84.2 ± 10.4%). Among demographical and clinical parameters, the only two overcoming 80% were age (81.7 ± 11.6%) and physiological cost index of walking (80.2 ± 12.6%). Then, the other four parameters closer to or higher than 80% were the forces of the quadriceps (paretic: 79.9 ± 11.5%; non paretic: 83.9 ± 10.1%) and those of the iliopsoas (paretic: 81.2 ± 6.7%, non-paretic: 88.7 ± 5.7%).

The ranges of motion of the joints of the paretic side weighted in a range between 44% (for hip adduction) and 73% (for hip abduction), with 64% for hip flexion, approximately 62% for ankle plantaflexion and knee flexion, and 51% for ankle dorsiflexion.

The distance covered during the 6 MWT had a weight of 72% for predicting the gait ratio. Cardiological parameters had baseline weights between 51% and 76%, and after 6 MWT between 61% and 73%. All clinical scale scores had a percentage weight of approximately 75%. The less-weighted factor was the gender of the patient (26 ± 10%).

The reliability of the neural network results was analyzed by the coefficient of variation. The standard deviation among the ten analyses ranged from 6–8% of the relevant means for the iliopsoas forces, 7–11% for walking speed, 12–14% for quadriceps forces, up to 30% for hip adduction ROM, and 37% for the patient’s gender.

Figure 3 shows the clear relationships of gait ratio with walking speed and stride length (the R^2^ of bi-exponential fit was 0.56 and 0.33, respectively). Interestingly, the constant asymptote for these fits were 1.658 and 1.616, respectively, which are very close to the value of golden ratio (about 1.618). Conversely, Figure 4 shows how the predictability of GR starting from muscle forces is not simple when classical fitting models are applied, even if they are not linear. Some data values are close to the golden ratio, but the coefficient of determinations R^2^ of the fits had low values (ranging from to 0.06 to 0.33). The complexity of these relationships was highlighted in Figure 5, which shows gait ratio values lower than the golden ratio for blue areas and higher for yellow areas, with the green regions representing a gait ratio closer to the physiological one.

## 4. Discussion

The main result of this study was the finding of a relationship between muscle forces and the ratio between stride and stance durations. In fact, the relationship of gait ratio with other spatio-temporal parameters of walking was expected, being also reported in previous studies [3,20,25]. Regarding these parameters, the interesting result was that the gait ratio approaches the physiological values of 1.618 for less affected patients walking with a long stride length and faster speed, as shown by the values of the asymptotes reported in Figure 3. Additionally, more simple analyses could highlight these relationships: even if non-linear, they are in fact monotonic over the entire range of values. As shown in Figure 4, the relationships between muscle forces and gait ratio are not monotonic, showing maxima or minima, in which the sign of the relationship is inverted. However, the artificial neural network associated high weight with these parameters for determining the gait ratio. For symmetric walking, the gait ratios are similar for both legs, but this is not the case with stroke patients. In this study, we focused our analysis on the gait ratio of the paretic limb, because we aimed to identify the factors influencing this potential target of neurorehabilitation. However, the gait ratio of the other limb could also be altered, whether directly by sensorimotor deficits or indirectly by complex strategies put in action to compensate for the limited movements of the paretic side.

During physiological walking, the iliopsoas is activated after the contralateral foot strike and achieves a maximum after the toe-off, then its activity is progressively reduced, disappearing at 80% of the gait cycle. The quadriceps are active during the two double support phase, and hence also from 50% to 62% of the gait cycle. Other muscles, such as gluteus maximus, hamstrings, gastrocnemius or soleus are not particularly active at the toe-off; only the tibialis anterior starts to be active again, but only to dorsiflex the feet for avoiding stumbles [4].

In a previous study of our group, we suggested that, for stroke patients, the alteration of gait ratio could be an effect, and not a cause, of speed reduction [3]. Independently, by the specific type of pathology, many patients showed a slower walking associated with (or often caused by) reduced ranges of motion of the lower limb joints, shorter strides, longer duration of the gait cycle, and a percentage increment of stance phase duration with respect to swing phase. The pendular mechanism of gait exploits inertia and gravity for generating a movement of body segments embedded into a specific harmonic structure, and slower speed reduced the possibility to exploit inertial forces [3,26,27]. The idea that gait ratio was related to the pendular mechanism of walking was already suggested in another article showing the importance of the fact that both anthropometric proportions and gait phase proportions are both related to the golden ratio [28]. In healthy subjects, muscle activations intervene at discrete times to drive the pendular oscillations of the system for compensating for the small loss of energy [4]. In patients with Parkinson’s disease the timing of internal cues for properly temporizing the muscle activations seemed to be altered [13], and external acoustic stimuli based on the golden ratio resulted effectively in an online restoring of a more harmonic walking [29]. Conversely, in stroke patients the problem could be not related to an alteration of internal timing, but on the alteration of muscle forces, that could be too weak or also hypertonic in case of spasticity [30], with a longer duration of the double support phase associated with less energy loss with speed [31].

Being the quadriceps active during the double support phases, the two quadriceps are simultaneously active during the foot-off that divides the stance phase and the swing phase [4]. In stroke patients, quadricep muscle forces exerted by the non-paretic limb are usually higher than those of the paretic side, and also than the quadricep muscles of healthy people, as a possible compensatory strategy to better support body weight and properly adjust the center of mass forward [32]. This may explain the reasons for which we found higher weight associated with muscle of the non-paretic side than with those of the paretic side: the compensation strategy relies on non-affected side muscles. Furthermore, it should also be noted that most of the patients enrolled in this study were older than 65 years old, with adjunctive possible effects of aging on their muscles.

The role of quadricep muscles are also in line with those of a study reporting that the functional electric stimulation of the tibialis anterior in stroke patients was more effective when combined with stimulation of the quadriceps than with stimulation of gluteus maximus for increasing walking speed and step length [33]. Not less important seemed to be the iliopsoas, which is a flexor of the hip and/or of the trunk and one of the most prominent hip muscles [34]. Figure 5 suggested the need for an equilibrium between the forces of these muscles to obtain harmonic walking. The subplot of the non-paretic side showed the obtainment of a gait ratio close to the golden ratio; if the quadriceps’ force ranges between 40 and 60 N, iliopsoas should also be in the same range; if the quadriceps range between 100 and 120 N, the iliopsoas should range between 70 and 90 N. The other subplot related to the paretic side showed two possible patterns for obtaining a golden gait ratio: if the paretic quadriceps range between 40 and 110, the same should be the iliopsoas; if the quadriceps have a force higher than 100 N, the iliopsoas should be able to develop a force higher than 60 N.

Finally, the weight of the association with the physiological cost index seemed in line with the results of Serrao and colleagues, who found that walking with a stride to stance ratio close to the golden ratio minimizes the energy expenditure related to walking, making gait an efficient mechanism from a metabolic point of view [14], according to an integrated model of locomotion [35]. These results were also confirmed by a recent article linking the golden ratio with the heart and locomotor rhythms coupling [36].

## 5. Conclusions

The Artificial Neural Network highlighted a weighted importance in determining the ratio between stride and stance durations for step length and walking speed, as expected, but also of hip muscle forces, more than of ranges of joint motion. It suggested a complex relationship, also related to possible compensation strategies, probably related to the pendular mechanism of walking that exploits well synchronized muscle activations. However, we also reported the high variability in the results of artificial neural networks when it repeatedly analyzed the data. However, the reliability seemed to be lower for the factors with a reduced weight, whereas it was higher for hip muscle forces, confirming their importance in determining the harmony of walking.

## Figures and Tables

**Figure 1 sensors-22-01374-f001:**
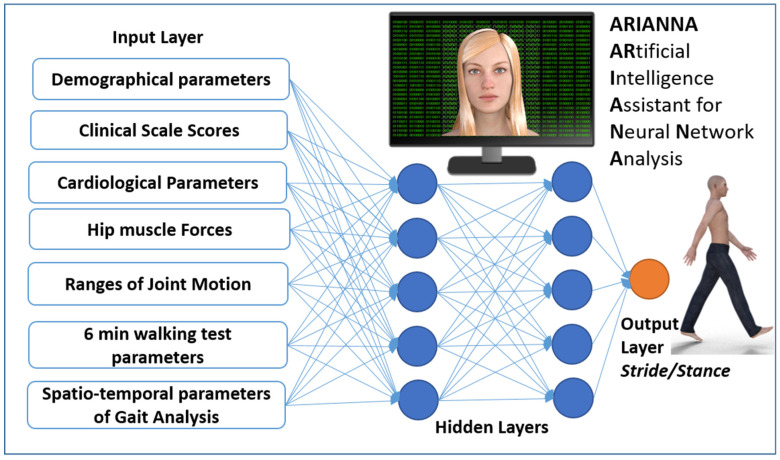
Schematic representation of the Artificial Neural Network used in this study.

**Figure 2 sensors-22-01374-f002:**
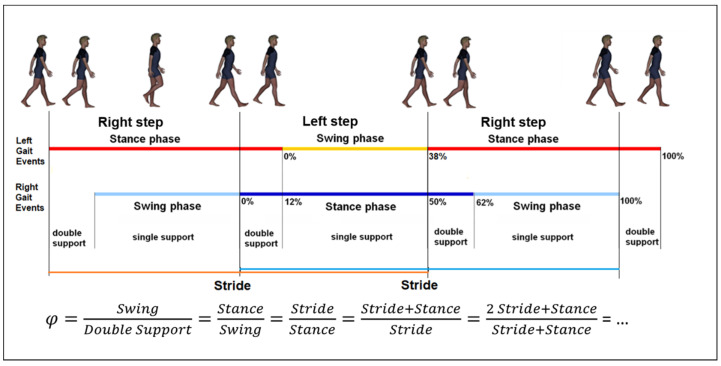
Schematic representation of the Gait Cycle and the equation of autosimilarity, defined as when the proportion between stride and stance durations is equal to the golden ratio (*φ*).

**Figure 3 sensors-22-01374-f003:**
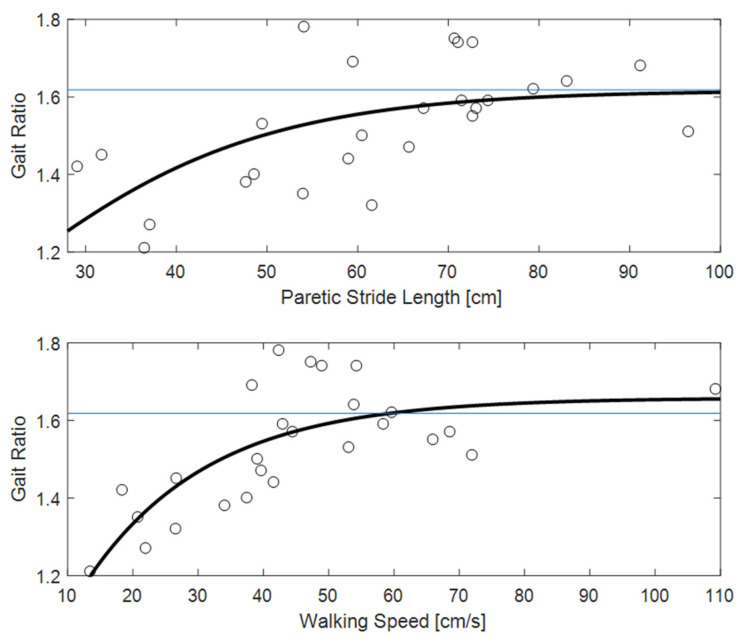
The gait ratio (stride duration/stance duration) reported with respect to paretic stride length (**top**) and walking speed (**bottom**). The horizontal blue line represents the theoretical value of the golden ratio (approximately 1.618); the wide black line is a bioexponential fit of the data.

**Figure 4 sensors-22-01374-f004:**
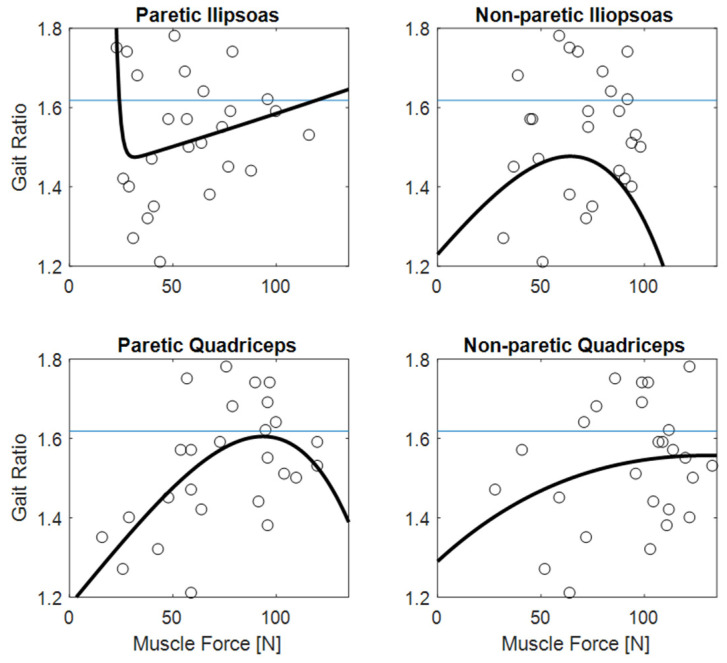
The gait ratio (stride duration/stance duration) reported with respect to paretic and non-paretic muscle forces. The horizontal blue lines represent the theoretical value of the golden ratio (about 1.618); the wide black bioexponential lines poorly fitted the data.

**Figure 5 sensors-22-01374-f005:**
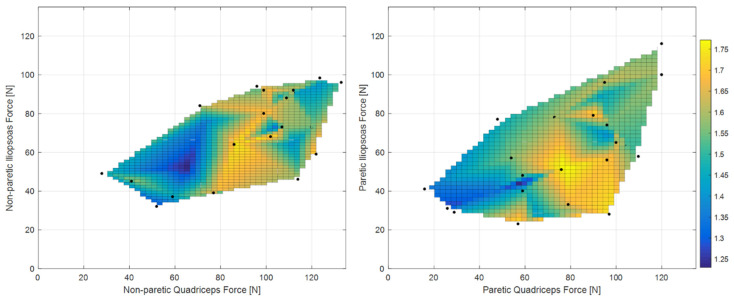
Colored map of gait ratio with respect to muscle forces obtained with surface fitting. Blue areas: gait ratio << golden ratio; green areas: gait ratio close to the golden ratio; yellow areas: gait ratio >> golden ratio (equal to 1.618).

## Data Availability

Data will be available on request to the corresponding author.

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
