# Peer review of "Artificial Neural Network Detects Hip Muscle Forces as Determinant for Harmonic Walking in People after Stroke"

_sensors, 2022, doi:10.3390/s22041374_

Round 1
Reviewer 1 Report
Artificial Neural Network detects hip muscle forces as determinant for harmonic walking in people after stroke
The study sought to understand the factors affecting the ratio of stride time to support time, as well as between stance/swing and between swing/double stance times in strokes. The strategy used by the authors was with the application of an artificial neural network. The paper is well-written and organized. Further, the study is interesting but several adjustments are needed before the article can be approved. The introduction lacks a rationale more related to the problem of the study. The background on neural networks, epidemiological data on strokes and the golden ratio is well done, but it is important to include literature that points out what is known about two fundamental points for the rationale of the study: 1) on the most important changes in gait mechanics in strokes, especially on spatiotemporal changes in gait (PMID: 29248191 and . 2) about the pendulum mechanism since the change in golden ratio is exactly impacted on the fundamental minimizing mechanism of gait, that of the inverted pendulum (PMID: 32694152 and PMID: 34197674 ).
Line 89 – range of joint motion (please revise in figure and remaining text).
Lines 89-91 – consider including the data of device (model, city, country) and the repeatability/validity of all procedures (references). For example consider using the following work (if suitable): PMID: 29400674
Line 102 – please, explain IOR . Also, measurements.
Lines 194-208 – The paragraph is very interesting but I miss original works confirming the inertial properties from the pendulum-like mechanism. Consider including some reference confirming these properties (e.g. PMID: PMID: 1011078, PMID: 24102934) even in strokes subjecs (e.g., PMID: 31220691).
Lines 206-208 – even been a basic information, consider including references supporting these statements (weak force, hypertonic/espasticity)
in figure 2 and in the other figures and results concerning the fitting curves of the data: consider including the standard error of the estimate.
Line 232 – consider using a reference to support this statement on efficiency (PMID: 30618802).
Reviewer 2 Report
The work of the authors «Artificial Neural Network detects hip muscle forces as determinant for harmonic walking in people after stroke» is devoted to the analysis of the timing of the walking cycle, which were previously discovered by this group and defined as the golden ratio. This paper investigates what patterns are manifested in other parameters of the biomechanics of walking in patients who have had a stroke. In general, this topic seems to be very interesting and rather unusual genre.
However, there are some inaccuracies in the work and a number of data in the results seem to me to be missing.
The line is 55 after “Artificial neural networks” (first mention) it is necessary to put abbreviation (ANN), which is used below.
The line 99 the word is “sterEophotogrammetric” one letter was missed.
Lines of 142-144. This proposal raises the question of which side are you talking about? Paretic, healthy, or is it something in between?
Lines 153-154. The question is that stride length is differentiated according to which side: paretic or healthy. And it can be significantly different. What is the length of the stride you mean here and why not for each side?
Lines 194-195. This is of course true. However, in my opinion, a decrease in walking speed is perhaps the most frequently encountered symptom. A lot of other changes: a change in the duration of the stance phase, an increase in the duration of the walking cycle, a decrease in the range of motion in the joints .... These are not symptoms of a specific disease. This is a slow walking syndrome. And it is necessary to separate this syndrome from specific changes for a particular pathology.
I would combine a significant part of the questions into this option. The work considers where one side (paretic), where two (paretic and healthy). But any biomechanical and EMG parameters in patients with hemiparesis as a result of stroke change for both sides. In this regard, it seems necessary to consider the parameters for each side. Except, of course, those that are outside (like speed, cadence...). In your previous work [19] you showed that, say, in terms of the duration of the stance phase, this category of patients may have asymmetry. And indeed it is. At the same time, when compensation is possible, then on the paresis side, the normative ratio of the duration of stance and swing phases will be maintained, but due to the fact that the stance phase will grow significantly on the healthy side. In any case, the asymmetry in such basic parameters as: walking cycle, stance phase, stride (for each side) length - will be abnormal. Asymmetric parameters of the walking cycle (especially stance phase) will not allow obtaining a good quality of walking, although an ideal ratio is quite possible if only the parameter of the paretic side is taken into account.
Reviewer 3 Report
The manuscript entitled “Artificial neural network detects hip muscle forces as determinant for harmonic walking in people after stroke” is a work which investigates the factors associated to the gait ratio, analyzing data by means of an artificial neural network. This works fits within the scope of Sensors. The figures of the manuscript are well presented and of high quality. Overall, the present manuscript presents an in-depth, but not an innovative work in terms of methodology experimental work. If the authors will clarify the listed points as below, the manuscript could be accepted.
- Title, 'Artificial Neural Network' to 'Artificial neural network'.
- Does the sample size is big enough (16 males and 10 females, 16 with left body hemiparesis and 10 with right hemiparesis)?
- It is better to put the figure caption under the figure.
- It would be nice the authors would add more discussion about the uniqueness of golden ratio (1.618).
Reviewer 4 Report
The manuscript proposes a method to associate different parameters with the gait ratio on stroke patients for gait analysis. The proposal is interesting, in particular, to be conducted with patients with stroke with a variety of biomarkers from different sources. The discussion is detailed and highlights the relevance of the work. However, there are some concerns that should be addressed in order to improve the understanding of the study.
Methods
It is mentioned a total of 33 parameters as input for the ANN. However, in Methods is a lack of which are the parameters from each one of the input groups shown in Figure 1. For example, gait cycle parameters were presented in section Results but they should be included previously in Methods.
The introduction explains different ways to obtain the gait ratio but it was not clear in the manuscript the one used for the study. Indeed, in the abstract, it was mentioned but it was not included in the Methods.
How was obtained the percentage of maximum importance used for analysis?
Discussion
The subjects involved in the study were mostly older adults, any discussion about this factor was included.
Round 2
Reviewer 1 Report
Congrats, the modifications were performed accordingly. The paper is ready for publishing. bw, reviewer.
Reviewer 2 Report
I read the detailed answers of the co-authors and the new editionof the article. In this form, this work satisfies me quite well.
Reviewer 4 Report
I thank the authors for their resposes, the manuscript has been improved. I agree with reviewed version and their publication.